# Ecological Relationships Between Woody Species Diversity and Propagation Strategies of *Aulonemia queko*

**DOI:** 10.3390/plants14050744

**Published:** 2025-03-01

**Authors:** Hugo Cedillo, Luis G. García-Montero, Fernando Bermúdez, Andrés Arciniegas, Mélida Rocano, Oswaldo Jadán

**Affiliations:** 1Centro Para la Conservación de la Biodiversidad y el Desarrollo Sostenible (CBDS), E.T.S.I. Montes, Forestal y del Medio Natural, Universidad Politécnica de Madrid (UPM), 28040 Madrid, Spain; luisgonzaga.garcia@upm.es; 2Grupo de Ecología Forestal, Agroecosistemas y Silvopasturas en Sistemas Ganaderos, Facultad de Ciencias Agropecuarias, Universidad de Cuenca, Cuenca 010114, Ecuador; fernando.bermudez@ucuenca.edu.ec (F.B.); andres.arciniegas@ucuenca.edu.ec (A.A.); melida.rocano@ucuenca.edu.ec (M.R.)

**Keywords:** floristic composition, diversity, density, *Aulonemia queko*, propagation, shoots, auxins, substrates

## Abstract

This study explores how floristic composition, diversity, and woody vegetation structure vary across floristic zones in Andean montane forests under the dominance of *Aulonemia queko* Goudot (Poaceae, Bambusoideae) dominance. As a culturally and ecologically significant non-timber forest product, *A. queko* plays a key role in shaping plant communities and requires effective propagation strategies for sustainable management. Significant differences in floristic composition were observed among zones, with indicator species identified in the lower and upper zones. However, despite environmental variability, species richness and structural attributes remained stable across the elevation gradient, suggesting resilience in woody plant communities. *A. queko* density was highest in the upper zone, while its basal area peaked in the lower and middle zones, probably shaping floristic composition through competitive interactions and habitat modification. Propagation experiments revealed that shoots with rhizomes exhibited higher survival and growth, particularly in mulch substrates with 1000 ppm indole-3-butyric acid (IBA), highlighting the importance of shoot type, substrate, and hormone dose. These findings suggest that *A. queko* is a structuring species and a potential restoration target. However, its dominance may alter forest composition, requiring adaptive management strategies that balance its ecological role with conservation and sustainable use, ensuring biodiversity and ecosystem resilience.

## 1. Introduction

Andean montane forests, extending from Venezuela to Argentina at elevations between 1200 and 3600 m above sea level, are globally recognized for their remarkable taxonomic, functional, and structural diversity [1,2]. These forests cover over 10% of the world’s forested area, hosting high levels of endemism and floristic richness while playing a critical role in climate regulation and ecosystem services [3,4]. However, human-induced threats, such as deforestation, land-use changes, and infrastructure development, pose significant challenges to their conservation [5,6]. Among these factors, the dominance of bamboo species has emerged as a key element influencing the montane forests’ structure, composition, and resilience [7,8,9].

Bamboo dominance profoundly shapes forest dynamics, impacting taxonomic and functional diversity by altering nutrient cycling, light availability, and resource competition [10,11,12]. Bamboo species enhance nutrient cycling through high leaf litter production and rapid decomposition, accelerating nutrient return to the soil and causing organic matter accumulation with a low carbon-to-nitrogen ratio [13,14]. Its dense, low growth form restricts juvenile tree regeneration, while cyclical phenology causes canopy fluctuations that create temporary gaps favoring pioneer species [7,9,15]. Additionally, its extensive root system intensifies competition for water and nutrients, reducing available space and lowering tree density and basal area in bamboo-dominated forests [10,16]. Environmental variations along altitudinal gradients further influence bamboo distribution and interactions with other species, altering canopy structure and forest biomass [15]. In mature and secondary montane forests, adaptive functional traits of woody species—such as wood density, root-to-shoot ratio, and specific leaf area—drive succession, ecological restoration, and ecosystem recovery by shaping growth dynamics and resource allocation [17,18]. Leaf morphology, including stomatal density and xylem structure, regulates water-use efficiency and photosynthetic capacity, which are critical for survival in fluctuating montane environments [19]. These dynamics highlight bamboo’s dual role as both a driver of forest composition and a key factor shaping ecosystem structure, emphasizing the need for management strategies that balance its expansion with biodiversity conservation and ecosystem functionality [15].

Among the diverse bamboo species, *Aulonemia queko* Goudot (Poaceae, the grass family; the common name is Queko bamboo) [20] holds ecological and socioeconomic importance in the Andean montane regions where it is endemic [21,22,23]. Although less studied than other economically significant bamboos, such as *Guadua angustifolia* or *Phyllostachys* spp., *A. queko* plays a critical role in local economies through its use in rural roofing and construction, handicrafts, Andean musical instruments, medicine, food, and forage production [24,25,26]. It also provides a vital habitat for species like the chusquera dove (*Paraclaravis mondetoura*), emphasizing its ecological significance [27]. Despite its potential, sustainable management and propagation of *A. queko* remain underexplored in Ecuador [28].

Globally, bamboo is integrated into sustainable management programs due to its role in poverty alleviation and its renewable resource potential [29,30]. However, the sporadic flowering cycle of *A. queko* (every 25–30 years) limits natural regeneration, making asexual propagation essential for its conservation and sustainable use [31,32,33]. Rhizome transplantation has been widely adopted in bamboo species due to its ability to store energy reserves and facilitate the emergence of new shoots, ensuring higher survival rates [34,35]. Similarly, the application of indole-3-butyric acid (IBA) has been extensively documented as a growth regulator that stimulates adventitious root formation and improves seedling establishment [34,36]. While no specific studies exist on the propagation of *A. queko*, these techniques have proven effective in species with similar morphological and ecological traits, such as *Bambusa balcooa*, *Bambusa vulgaris*, *Dendrocalamus asper*, and *Dendrocalamus latiflorus* [35,36,37,38,39]. Thus, adapting these methods to *A. queko* is a viable strategy to enhance its propagation, ensuring long-term sustainability and conservation.

This study aims to contribute to the conservation and management of *A. queko* by addressing three objectives: (1) analyzing variations in the composition, diversity, and structure of woody vegetation along an altitudinal gradient; (2) examining the relationship between the dominance (density and basal area) of *A. queko* and the diversity and structure of woody plants; and (3) evaluating survival and morphological growth parameters of *A. queko* using different propagation techniques. Specifically, this study investigates whether higher survival rates and enhanced growth are achieved in shoots with rhizomes using organic-enriched substrates and higher doses of indole-3-butyric acid (IBA).

## 2. Results

### 2.1. Floristic Composition

The PERMANOVA analysis revealed significant variation in floristic composition among the floristic zones (*p* = 0.009). The pairwise comparison revealed significant differences in floristic composition only between the lower and upper floristic zones (*p* = 0.027). In comparation, no significant variations were detected between the lower and intermediate (*p* = 0.167) or intermediate and upper zones (*p* = 0.179). These patterns were reflected in the NMDS plot (stress = 0.19, Figure 1), where the first axis explained 57.79% of the variation, while the separation along the second axis was less relevant, accounting for only 42.21%. Two indicator species were identified in the lower floristic zone, none were identified in the intermediate zone, and one was identified in the upper zone.

### 2.2. Composition, Diversity, Structure of Woody Species, and Dominance of A. queko

The richness of woody species did not show significant differences between the floristic zones (*p* = 0.399). Similarly, the structure of woody species, assessed by abundance (*p* = 0.521) and basal area (*p* = 0.8788), also did not exhibit significant variations. In contrast, the dominance through the density of *A. queko* showed significant differences (*p* < 0.0001), being higher in the upper floristic zone and lower in the lower zone (Figure 2A). Basal area was significantly greater (*p* < 0.0001) in the lower and intermediate zones compared to the upper zone (Figure 2B).

### 2.3. Relationships Between Ecological Parameters of Woody Species and A. queko

Although none of the abundance and basal area variables of *A. queko* showed significant correlations with the richness and structure variables of woody vegetation, the basal area of *A. queko* and elevation explained the variation in the floristic composition of woody species (Table 1).

### 2.4. Survival of A. queko

The survival of *A. queko* was significantly higher in shoots with rhizome (Figure 3; *p* < 0.0001). However, this parameter did not show significant differences in substrate factors (*p* = 0.08814), hormone doses (*p* = 0.54652), and in the interactions between these factors. The variables related to the diameter growth of the first stem, root number, and specific leaf area (SLA) showed no correlations; therefore, their results will be shown below. Correlations among all response variables are presented in Appendix A.

### 2.5. Growth in Diameter

Diameter growth was significantly higher in rhizome-derived shoots (Figure 4A; *p* < 0.0001). It was also more significant in the mulch substrate (Figure 4B; *p* = 0.0232) and when the hormonal dose B (1000 ppm) was applied (Figure 4C; *p* = 0.0109). Furthermore, diameter growth was statistically significant when considering the interactions among the three evaluated factors (shoot type, substrate, and hormonal dose), highlighting their interdependence (Figure 4D). Here, the highest diameter value was observed in interaction b (Figure 4D, Table 2), corresponding to the combination of culm with rhizome shoots, mulch substrate, and the hormonal dose B (1000 ppm) of IBA.

The root number was significantly higher in shoots with rhizomes (*p* = 0.0004, Figure 5A). This parameter was also statistically significant (*p* = 0.0021, Figure 5B) when considering the interactions among the three evaluated factors (shoots, substrate, and dose), highlighting their interdependence. The highest values were observed in interaction k, corresponding to shoots with rhizomes, peat substrate, and applying hormone C (3000 ppm of IBA, Table 2).

The specific leaf area (SLA) was significantly higher (*p* < 0.0001) in shoots without rhizomes (Figure 6A), with black soil (*p* < 0.0001; Figure 6B), and when the hormone dose C (3000 ppm of IBA) was applied (*p* = 0.01; Figure 6C). It was also significant under the interaction between shoots and substrate, being highest in the interaction d (Table 3), specifically in shoots without rhizomes and peat as substrate (*p* = 0.036; Figure 6D).

## 3. Discussion

This study analyzed the variation in floristic composition, diversity, and structure of woody vegetation along an altitudinal gradient, focusing on the dominance of *A. queko*. The results registered significant differences in floristic composition between the lower and upper floristic zones, while species richness and structural parameters remained unchanged across elevations. Although *A. queko*’s density showed no direct correlation with woody vegetation diversity or structure, its basal area and elevation significantly influenced floristic composition, highlighting its role in shaping community dynamics. Additionally, this study evaluated the survival and morphological growth of *A. queko* under different propagation conditions, considering the use of rhizomes, mulch-enriched substrates, and the application of indole-3-butyric acid (IBA).

### 3.1. Variation in Composition, Diversity, and Structure Along the Elevation Gradient

The results revealed significant variation in floristic composition across floristic zones, as indicated by PERMANOVA and a significant correlation with elevation. These findings are consistent with the well-documented influence of elevation on plant community composition, driven by changes in abiotic factors, such as temperature, moisture, and soil properties [40,41]. Identifying indicator species in the lower and upper zones suggests that these zones provide distinct environmental conditions. In contrast, the absence of indicator species in the middle zone may reflect a transition zone where ecological conditions overlap, supporting a mixture of species from adjacent zones. Woody plants’ diversity (richness) and structural parameters (abundance and basal area) showed no significant variation along the elevation gradient. This consistency may reflect the resilience of woody plant communities in gradients with limited resources.

Environmental changes driven by elevation or stabilizing processes like seed dispersal and local competition may exert minimal influence on maintaining the diversity and structure of these communities. However, the density of *A. queko* was significantly higher in the upper zone, indicating its preference for colder, potentially less competitive conditions typical of higher elevations [42]. In contrast, the greater basal area observed in the lower and middle zones likely underscores its competitive advantage in resource-rich environments.

### 3.2. Influence of A. queko on Woody Vegetation Composition, Diversity, and Structure

Although the density and basal area of *A. queko* were not significantly correlated with woody vegetation diversity or structural parameters, its basal area explained the variations in floristic composition. This suggests that *A. queko* subtly influences community dynamics through resource competition or habitat modification rather than directly affecting overall diversity or structure. Similar effects have been observed in other bamboo species, such as *Guadua* spp., which impose competitive constraints on tree recruitment through high dominance and mechanical interference [8,9]. However, in secondary forests, *Aulonemia aristulata* has been associated with higher recruitment of tree and shrub seedlings in bamboo-dominated plots compared to non-bamboo-dominated ones [11]. These contrasting effects likely arise from differences in bamboo species traits and forest successional stages. While *Guadua* species in mature forests restrict tree recruitment through intense competition and mechanical suppression, *A. aristulata* in secondary forests facilitates seedling establishment by enhancing light availability, improving soil structure, and increasing microsite heterogeneity as succession advances following its natural die-off [11].

The role of *A. queko* as a dominant species (accord basal area) aligns with previous studies on bamboo as ecosystem engineers. For instance, *Guadua sarcocarpa* and *G. weberbaueri* modify light environments, alter soil nutrient availability, and create dense understories that limit tree regeneration [43]. Similarly, the high density of *A. queko* in the upper zone may enhance its ability to dominate these environments. At the same time, its greater basal area in the lower and middle zones likely reflects its impact on space and resource availability. These patterns reinforce the notion that bamboo dominance can drive shifts in floristic composition, supporting the idea that *A. queko* plays an important role in Andean montane forest structure. However, further research is needed to determine whether these effects are primarily driven by competitive exclusion, facilitation, or a combination of both mechanisms, as observed in other bamboo-dominated ecosystems.

### 3.3. Survival and Influence of Rhizomes

Our findings indicated a significantly higher survival rate in shoots with rhizomes, consistent with previous studies demonstrating the higher survival of plants propagated with rhizomes [44]. Rhizomes provide an additional energy reserve, facilitating the establishment of new shoots under suboptimal conditions [45]. However, while rhizome-based propagation resulted in higher survival, the type of substrate (mulch) and the hormonal treatment (IBA) did not produce significant differences in survival rates. This lack of significance could be attributed to multiple factors. One possibility is that the sample size was insufficient to detect subtle differences in survival across treatments, suggesting that larger-scale trials may be necessary to refine these conclusions. Alternatively, the ecological traits of *A. queko* may reduce dependency on external growth stimulants when rhizomes are present, as they already provide the necessary resources for initial establishment. This fact aligns with previous research indicating that substrate and hormones play a more substantial role in post-establishment growth parameters, such as root development and shoot diameter, rather than initial survival [46]. Further research considering larger sample sizes, varying environmental conditions, and extended monitoring periods would help us clarify whether the limited impact of substrate and hormones results from methodological constraints or reflects a broader ecological adaptation of *A. queko.*

### 3.4. Diameter Growth and Interaction with Substrate and Hormone Doses

Diameter growth was significantly higher in rhizome-bearing shoots, with the most significant increase observed when combined with the mulch substrate and the lowest tested IBA dose (1000 ppm). This fact aligns with research on other species, where hormones, particularly IBA, enhance root and shoot growth, especially when paired with suitable substrates [47]. Mulch improves soil moisture retention, reduces evaporation, and maintains a stable microclimate, which benefits propagules in early establishment stages by mitigating drought stress [48,49]. As it decomposes, mulch contributes organic matter, enhancing nutrient availability and potentially stimulating root proliferation [50]. Additionally, it promotes beneficial microbial communities, including mycorrhizal fungi and plant growth-promoting bacteria, which facilitate nutrient uptake and improve plant vigor [44]. The interaction between substrate, rhizome presence, and IBA highlights the importance of optimizing these factors to support both root initiation and long-term growth [51].

Similar patterns have been observed in other bamboo species. In *Dendrocalamus giganteus* Wall. ex Munro, Kalanzi and Mwanja [52] found that lower IBA doses (0.6%) were more effective in stimulating sprouting and rooting, which aligns with the present study, where the lowest tested dose (1000 ppm) yielded the best results. This finding is consistent with Davies [53], who emphasized that plant hormones are more effective at low concentrations, while excessive doses can be toxic. High IBA concentrations may induce increased ethylene synthesis, inhibiting regular auxin transport [54] and potentially damaging epidermal tissues [55]. Additionally, IBA has been shown to enhance root and shoot growth when combined with suitable substrates [47]. The role of mulch in improving soil moisture retention and nutrient availability, as highlighted by Hartmann [44], reinforces its importance in optimizing propagation success. These results collectively highlight the importance of balancing substrate, hormone concentration, and rhizome presence to enhance root initiation, nutrient uptake, and plant vigor.

### 3.5. Root Abundance and Interaction Effects

The number and quantity of roots were significantly higher in shoots with rhizomes, and specific interactions between shoots, substrate, and hormone doses further enhanced this effect. The interaction k (rhizomes, peat, and 3000 ppm IBA) resulted in the highest root number. Rhizomes serve as carbohydrate and hormone reservoirs, accelerating root differentiation and improving water and nutrient uptake efficiency, particularly when paired with peat, which enhances aeration and moisture retention [45,56]. In natural conditions, root system vigor is crucial for plant establishment, particularly in clonal bamboo species, where early root proliferation enhances competitive ability and resilience to environmental stress [57].

Additionally, the interaction between IBA and rhizomes probably stimulates auxin transport and cell elongation, promoting root primordia differentiation and increasing the efficiency of resource acquisition. Auxins play a key role in root development, regulating lateral root initiation through the modulation of transport and signaling genes, although this process varies between monocots and dicots. In dicots, lateral root initiation occurs in the pericycle at the xylem pole, where auxin accumulation activates key regulators that promote cell division and the differentiation of root primordia [58]. In contrast, this process is more complex in monocots such as *A. queko* and others, involving both the pericycle and endodermal cells opposite the phloem [59]. These differences can be attributed to variability in the expression of proteins responsible for the polar transport of auxins through tissues, facilitating their hormonal distribution in the root system; in dicots, inhibition of these transporters drastically reduces lateral root formation, whereas in monocots, adventitious root formation is less dependent on these mechanisms [58,59]. Understanding these differences in auxin signaling can improve propagation strategies for species such as bamboo, where clonal regeneration largely depends on the hormonal balance within the rhizomes.

### 3.6. Specific Leaf Area (SLA) and Growth Parameters

The results show that specific leaf area (SLA) was significantly higher in shoots without rhizomes, suggesting a greater allocation of resources to leaf expansion rather than root development. This fact aligns with previous studies that indicate that shoots without rhizomes tend to prioritize aerial biomass, resulting in a higher SLA [60,61]. However, this finding is unexpected, as rhizomes typically enhance overall vigor by providing stored resources that support shoot and root development. One possible explanation is that the absence of rhizomes induces a stress-related morphological adaptation, where plants increase SLA as a compensatory strategy to maximize photosynthetic efficiency under resource-limited conditions, as described by Zhang [62]. Similar responses have been observed in other species, where higher SLA is associated with increased plasticity in response to physiological stress or competition for light [63].

Additionally, substrate type played a significant role, with black soil promoting higher SLA, which may be attributed to its higher organic content and improved nutrient and water retention. These factors have been shown to influence leaf morphology by modulating nutrient availability and water uptake [63]. The application of 3000 ppm IBA also increased SLA, likely due to its role in stimulating cell expansion and leaf development [47,64]. The interaction between shoots without rhizomes and peat substrate resulted in the highest SLA, highlighting the importance of the specific combination of shoot type and substrate in shaping leaf morphology. These findings are consistent with previous research that emphasizes the role of substrate and plant growth regulators in promoting plant development [65,66].

Overall, these results suggest that the higher SLA observed in shoots without rhizomes may not necessarily indicate greater vigor but rather a plasticity response to resource limitations. Future studies should explore whether this shift in biomass allocation has long-term effects on plant survival and competitiveness, particularly in natural environments where resource availability fluctuates.

## 4. Materials and Methods

### 4.1. Study Area

The floristic study (objectives one and two) was conducted in montane forests on the western slope of the Andes in Azuay and Guayas Provinces (Figure 7A), where *A. queko* naturally grows between 2000 and 3500 m of elevation. These secondary forests, verified by their floristic composition and structure, experience temperatures ranging from 6 °C to 28 °C and annual precipitation ranging between 800 and 1250 mm. The natural gradient of *A. queko* distribution was divided into three floristic zones based on elevation: Lower (2000–2500 m), Intermediate (2500–3000 m), and Upper (3000–3500 m). Five 50 × 20 m (0.1 ha) plots were randomly established in each zone, ensuring a minimum distance of 300 m between plots for data independence (Figure 7B). Sampling occurred between January 2021 and November 2022, following the floristic evaluation protocols for Andean native forests proposed by [67].

### 4.2. Research Area and Experimental Design for the Propagation of A. queko

A nursery was established at the Faculty of Agricultural Sciences to evaluate the propagation of *A. queko*. An experimental study was conducted using both descriptive and inferential analysis. This site is geographically located at 02°55′16′′ S and 079°01′30′′ W in Azuay Province at an elevation of 2596 m. The area experiences average annual temperatures ranging from 13 to 19 °C and average yearly precipitation of 950 mm. A completely randomized design (CRD) with a factorial arrangement was employed, evaluating three factors: (1) The vegetative part (shoots with and without rhizome), (2) the type of substrate (mulch, peat, and black soil), and (3) doses of indole-3-butyric acid (IBA) (0, 1000, 3000, and 5000 ppm). These factors resulted in 72 experimental units, containing six *A. queko* plants, for 432 shoots. Over the seven months of the experiment, plants were irrigated twice a week; manual weed control was implemented; and no fertilizers, pesticides, or herbicides were used. This experimental design enabled the evaluation of aerial and underground parameters under controlled conditions, ensuring reliable results.

### 4.3. Preparation of Propagation Material (First Factor)

The vegetative parts, or shoots with and without rhizomes, were collected from one-year-old plants (Figure 8A) from natural regeneration in secondary montane tropical forests where *A. queko* grows naturally. Following the recommendations of [68], these plants were then acclimatized for one year in the nursery, with a substrate consisting of mulch and soil from the same forest (Figure 8B). From these plants, those exhibiting desirable and homogeneous health and vegetative development (such as turgid leaves and stems) were selected (Figure 8C). The substrate was then carefully removed (Figure 8D). A single shoot was attached to the rhizome (Figure 8E), and 80% of the root system was pruned (Figure 8F). In the case of shoots without rhizomes (Figure 8G), the shoot was carefully separated from the mother plant. Subsequently, the leaves of both shoots were pruned by 30% to reduce evapotranspiration during the rooting process. In addition, the main culm of the plant was removed, since it was not part of the propagation test (Figure 8H).

### 4.4. Preparation of Substrate (Second Factor)

The substrates used in the experiment were mulch, black soil, and peat, all disinfected before use. Perforated plastic bags (45 cm × 60 cm) were filled with specific amounts of each substrate: 4 kg of mulch (576 kg in total), 6.4 kg of black soil (921.6 kg in total), and 2.8 kg of peat (403.2 kg in total). The mulch consisted of leaf litter collected from the forest, while the black soil was also sourced from the same environment. The peat was of the Kekkila brand, manufactured at Vantaa, Finland, a sterile natural compound renowned for its excellent water retention properties and a pH range of 5 to 7. Each shoot, whether with or without a rhizome, was planted at a depth of 3 to 4 cm within the substrate and bags, ensuring optimal conditions for growth and development.

### 4.5. Application of Indole-3-Butyric Acid (IBA) (Third Factor)

Indole-3-butyric acid (IBA) was used as a plant growth regulator at concentrations of (A) 0 ppm (control), (B) 1000 ppm, (C) 3000 ppm, and (D) 5000 ppm. The solvent used was 97% alcohol, and the IBA doses were precisely measured using an Adventurer Pro electronic balance, manufactured by OHAUS Corporation at the United States. The IBA was mixed with alcohol and then diluted in distilled water, with the resulting solutions stored in plastic containers. The rhizome shoots were immersed in the IBA solutions for 5 min and then exposed to a fan for 30 s to evaporate the volatile alcohol. Finally, the IBA-treated shoots were placed in bags containing different substrates [36].

### 4.6. Data Collection

For floristic inventory, within each plot, all woody plant individuals with a diameter at breast height (DBH) ≥ 2.5 cm were identified at the species level. The floristic composition was determined using the collected data, and key structural parameters, including density (number of individuals/ha) and basal area (m^2^/ ha), were quantified. A similar approach was used for *A. queko*, where dominance was assessed based on density and basal area. Density was estimated by counting the groups, while basal area was calculated for each culm within these groups. The SLA is calculated using the following formula:Basal area = 0.7854 × DBH^2^
where

DBH: Diameter at breast height.

The response variables in the propagation of *A. queko* included survival percentage, diameter growth of the first stem, height of the first shoot, number of leaves on the first shoot, length of the most developed root, root number, and specific leaf area (SLA). Survival was evaluated by counting the live and dead shoots in each experimental unit after seven months, from January to August 2022, after the experiment was established. The survival percentage was determined using the formula proposed by [69].Survival (%) = Lp/((Lp + Dp)) × 100
where

Lp: living plants.Dp: dead plants.

The diameter growth of the first stem was measured at the center of the first internode using a Vernier caliper. The height of the first shoot was measured from the base of the internode at the substrate level to the apex of the culm. Only green true leaves and pseudo-petiolate were included in the leaf count, while non-true leaves, such as basal, leathery, and brown ones, were excluded. The measurement and counting of parameters, including the length of the most developed root and root number, were performed at the end of the experiment, after 7 months. This period was chosen because it represents a critical and optimal stage for the species’ initial vegetative development, during which shoots, roots, and leaves grow significantly and remain stable. For this, six seedlings were randomly selected per experimental unit, totaling 144 seedlings. The seedlings were carefully removed from the plastic bags, and the roots were cut at the stem collar. The length of the longest root was measured, and the number of secondary roots on each plant was counted. The average of these parameters, calculated from the six seedlings in each experimental unit, was used as the representative value for each evaluated level combination. The specific leaf area (SLA) was calculated using the ratio of leaf area to leaf dry matter. Five leaves were collected from each experimental unit and dried at 80 °C for 48 h. These measurements assessed key parameters of vegetative development in *A. queko*. The formula for calculating the SLA is as follows:Specify leaf area (SLA) = (Leaf area)/(Dry matter)

### 4.7. Data Analysis

Variation in floristic composition was analyzed using a semiparametric permutational multivariate analysis of variance (PERMANOVA) [70]. Pairwise comparisons among floristic zones were conducted using permutation MANOVAs (*p* < 0.05) based on a distance matrix calculated with the Bray method. The composition of each floristic zone was subsequently visualized through a non-metric multidimensional scaling (NMDS) analysis. Indicator species within each floristic zone were identified using the R package ‘indspecies’ (from R Studio 4.2.3) [71]. Diversity and structural parameters were evaluated based on the woody species’ density and basal area. Variations in these parameters were analyzed using an analysis of variance (ANOVA), following the confirmation of normality and homoscedasticity assumptions. This method was also applied to assess the density and basal area of *A. queko*.

The relationship between the floristic composition of woody species and the density and basal area parameters of *A. queko* was analyzed using the envfit function from the ‘vegan’ package in R. Here, elevation was included as a predictor of woody composition, providing explanatory rigor through a key physiographic environmental variable. This tool allowed for the identification of *A. queko* variables significantly associated with the NMDS analysis axes.

A correlation analysis was initially conducted to identify and remove highly correlated response parameters (r ≥ 0.8) in order to analyze the parameters related to the survival and morphological growth of *A. queko* using the find Correlation function from the ‘caret’ package in R. [72]. This step minimized redundancies by excluding parameters with similar response patterns to the evaluated factors and interactions. Subsequently, data normality was assessed using the Shapiro–Wilk test. For datasets that did not meet normality assumptions, generalized linear models (GLMs) were fitted using distributions such as Negative Binomial, Gaussian, and Poisson. The best model was selected based on the lowest ratio of deviance to residual degrees of freedom. An analysis of variance (ANOVA) was then applied to the selected model using the chi-square test. For significant factors and interactions (*p* ≤ 0.05), post hoc tests were performed using Tukey’s test to identify significant differences between the levels of the evaluated factors and interactions.

## 5. Conclusions

The significant variation in floristic composition along the elevation gradient highlights the role of elevation in shaping plant communities, with indicator species unique to specific zones reflecting ecological adaptations. While species richness and structural metrics like basal area and abundance remained constant across zones, the density of *A. queko* exhibited a precise gradient, emphasizing its ecological prominence in the upper zone.

Although *A. queko*’s dominance (basal area) did not significantly correlate with richness or structural metrics, it played a pivotal role in explaining variations in floristic composition, suggesting its influence on community assembly. The lack of correlation between *A. queko* basal area and woody plant diversity underscores potential niche complementarity or coexistence mechanisms within these ecosystems.

Survival and growth parameters, particularly diameter and root number, were significantly enhanced in shoots with rhizomes, indicating their suitability for propagation efforts aimed at restoration or cultivation. Interactions among shoot type, substrate, and hormonal dose revealed complex interdependencies, with mulch substrate and a 3000 ppm IBA dose optimizing growth, emphasizing the need for integrated management practices for effective propagation.

## Figures and Tables

**Figure 1 plants-14-00744-f001:**
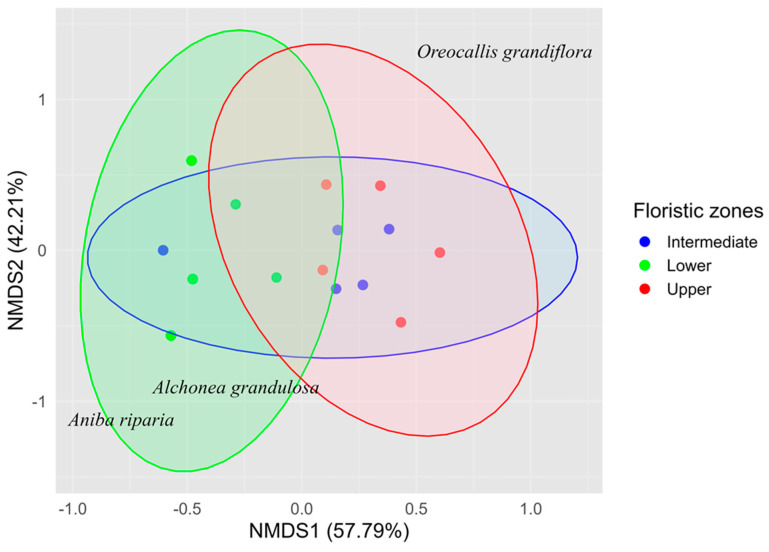
Non-metric multidimensional scaling (NMDS) plot illustrating spatial variation in floristic composition across the three floristic zones. Indicator species associated with each zone are also shown.

**Figure 2 plants-14-00744-f002:**
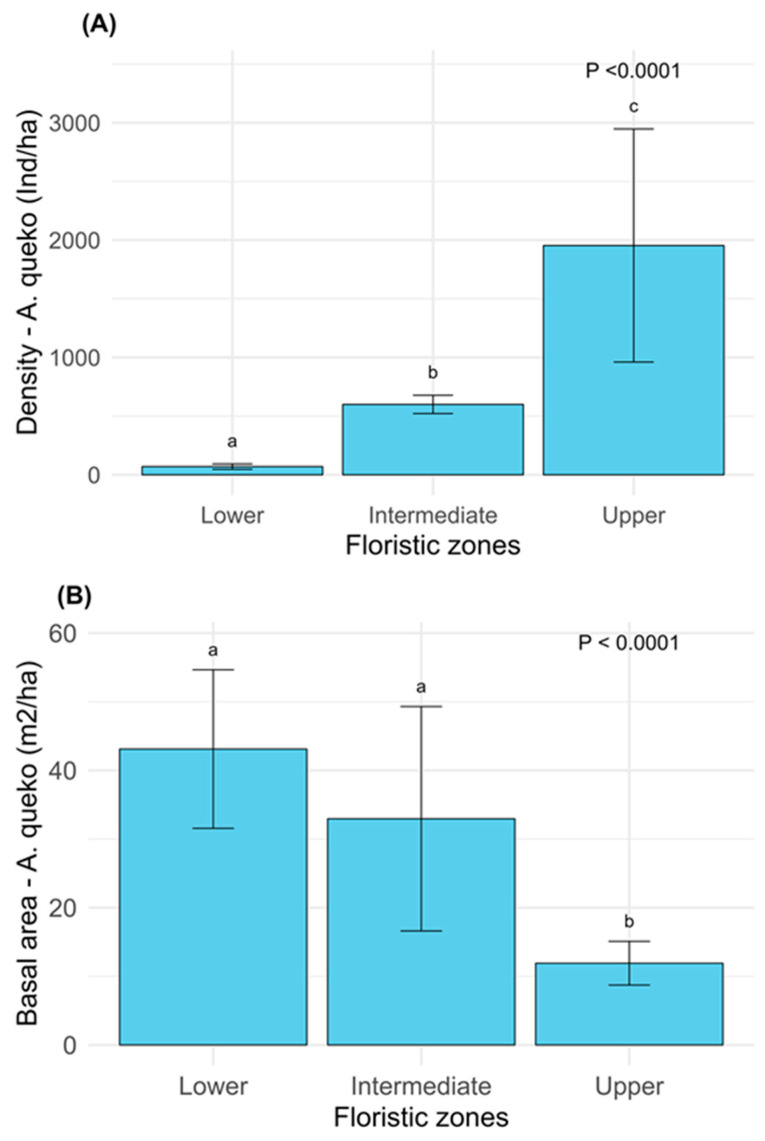
(**A**,**B**) Density and dominance (basal area) of *A. queko* shown in the three floristic zones. Different letters above the bars indicate significant differences (*p* ≤ 0.05).

**Figure 3 plants-14-00744-f003:**
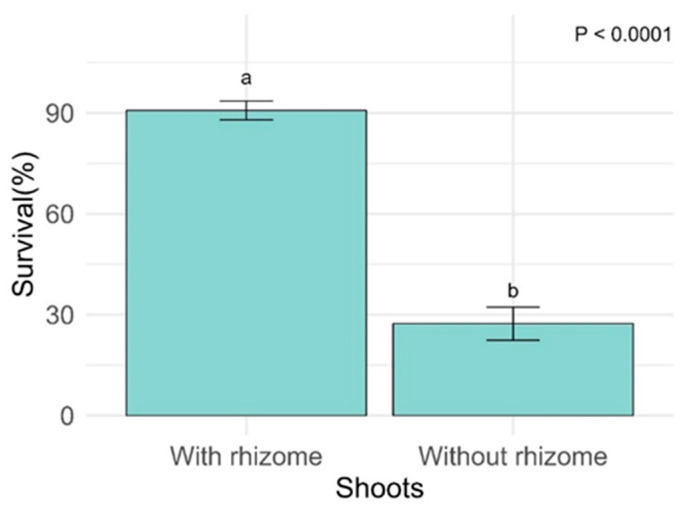
Survival of *A. queko* plants differentiated by shoots. Different letters indicate significant differences (*p* ≤ 0.05).

**Figure 4 plants-14-00744-f004:**
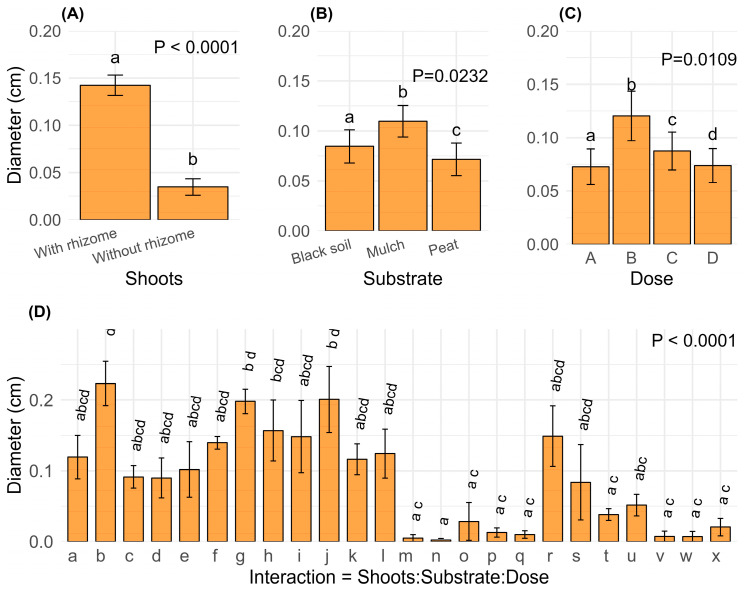
Means standard error of diameter growth obtained by ANOVA (GLM, Gaussian) for the significant factors: (**A**) shoots, (**B**) substrate, (**C**) dose; (**D**) interaction between shoots, substrate, and dose. Table 2 shows the meanings of the codes (letters) that correspond to the interactions between shoots, substrate, and hormone dose. Different letters above the bars indicate significant differences (*p* ≤ 0.05).

**Figure 5 plants-14-00744-f005:**
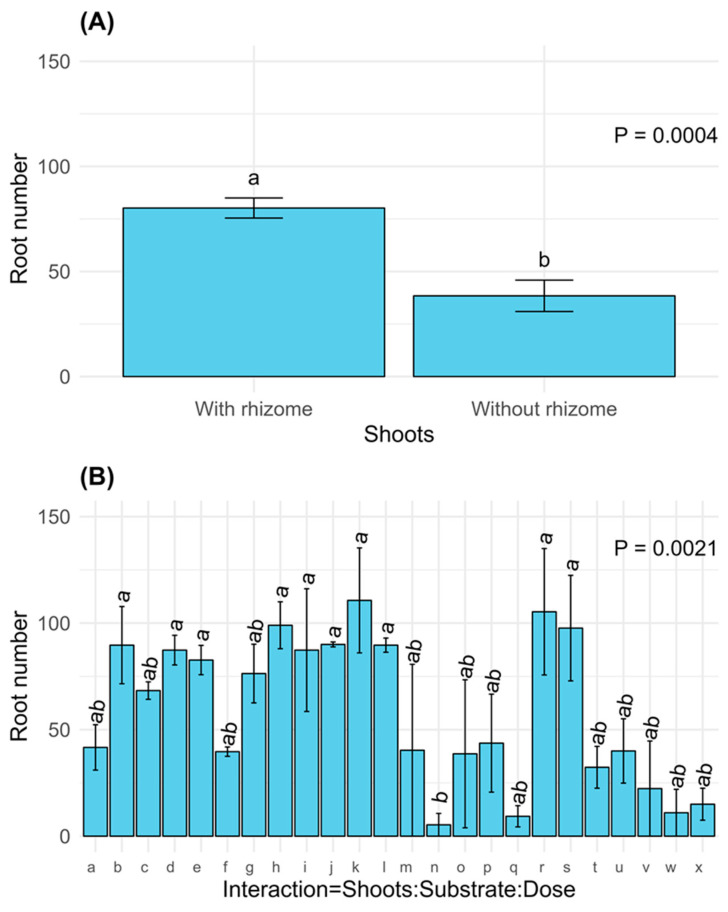
Means standard error of root number obtained by ANOVA (GLM, negative binomial, log link) for the significant factors: (**A**) shoots; (**B**) interaction between shoots, substrate, and dose. The meanings of the interaction codes (x-axis of Figure 5B) are shown in Table 2. Different letters above the bars indicate significant differences (*p* ≤ 0.05).

**Figure 6 plants-14-00744-f006:**
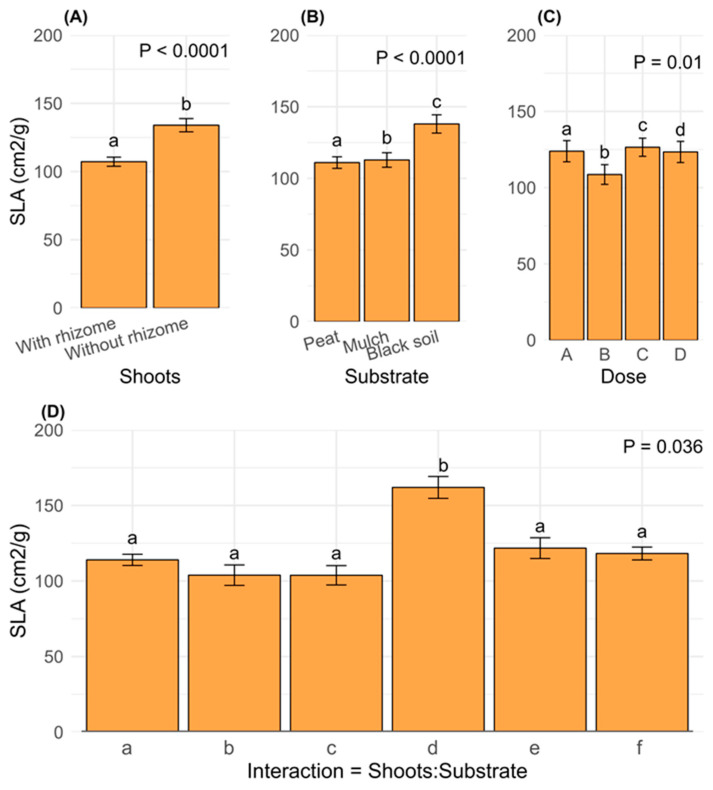
Means standard error of SLA by ANOVA (GLM, negative binomial, log link) for the significant factors: (**A**) shoots, (**B**) substrate, (**C**) dose; (**D**) interaction between shoots and substrate. Table 3 shows the meanings of the codes (letters) that correspond to the interactions between shoots and substrate. Different letters above the bars indicate significant differences (*p* ≤ 0.05).

**Figure 7 plants-14-00744-f007:**
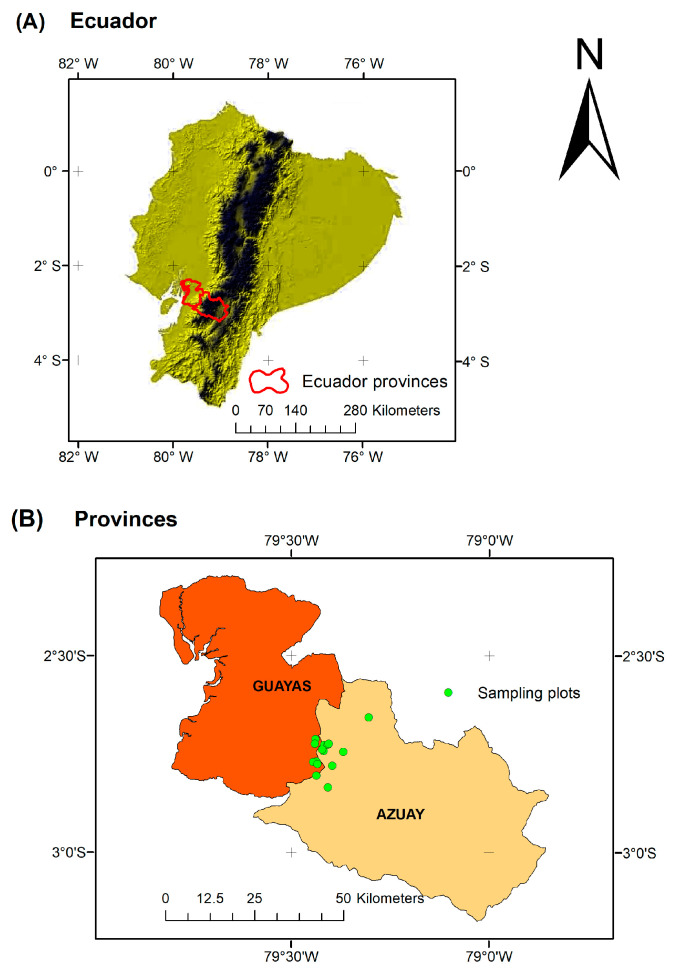
(**A**,**B**) Location map of the ecological study area. In (**A**), the black color represents the relief of the Andes Mountains, while dark yellow indicates areas below 1000 m in elevation.

**Figure 8 plants-14-00744-f008:**
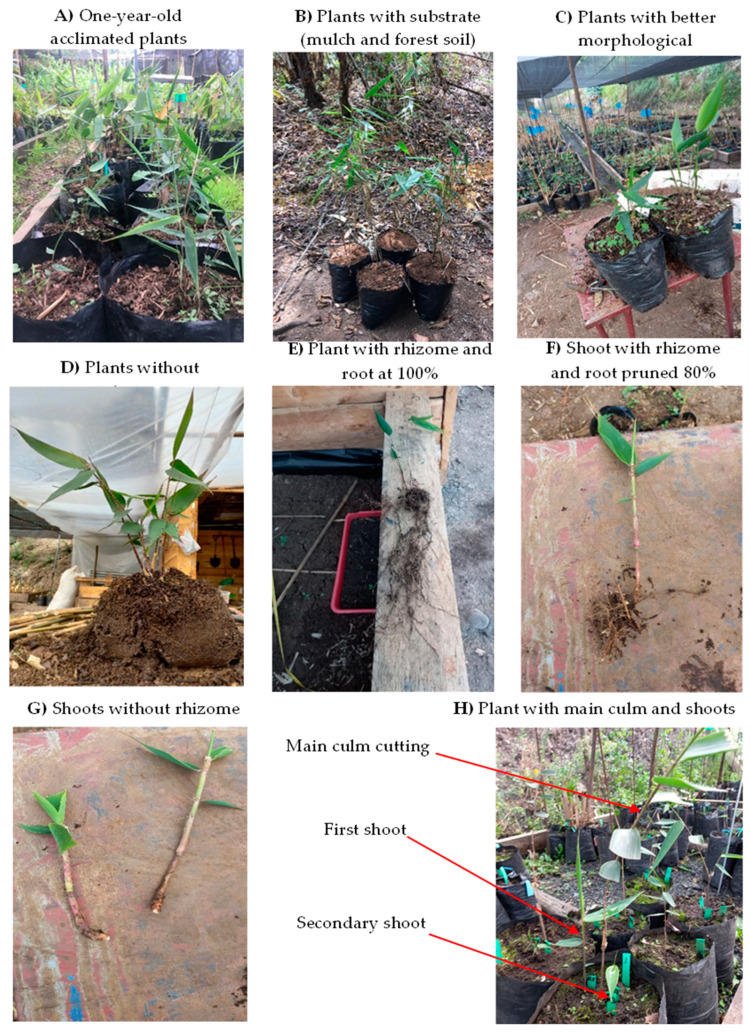
Selection, acclimatization, and extraction of shoots with rhizome and without rhizome of *A. queko*.

**Table 1 plants-14-00744-t001:** Density and basal area parameters of *A. queko* significant with NMDS ordination.

NMDS1	NMDS2	r2	Pr (>r)	*p*-Value
Density *A. queko*	0.57	0.82	0.31	0.118
Basal area *A. queko*	−0.94	−0.34	0.43	0.037 *
Elevation	0.99	0.12	0.66	0.001 ***

Signif.: *** < 0.001, * < 0.05.

**Table 2 plants-14-00744-t002:** Interaction code and corresponding evaluation factors for the different propagation parameters of *A. queko*. The dose description is described in the methodology.

Code Interaction (X Axis—Figure 4 and Figure 5)	Shoots	Substrate	Dose	Code Interaction (X Axis—Figure 4 and Figure 5)	Shoots	Substrate	Dose
a	With rhizome	Mulch	A	m	Without rhizome	Mulch	A
b	With rhizome	Mulch	B	n	Without rhizome	Mulch	B
c	With rhizome	Mulch	C	o	Without rhizome	Mulch	C
d	With rhizome	Mulch	D	p	Without rhizome	Mulch	D
e	With rhizome	Tnoire	A	q	Without rhizome	Tnoire	A
f	With rhizome	Tnoire	B	r	Without rhizome	Tnoire	B
g	With rhizome	Tnoire	C	s	Without rhizome	Tnoire	C
h	With rhizome	Tnoire	D	t	Without rhizome	Tnoire	D
i	With rhizome	Peat	A	u	Without rhizome	Peat	A
j	With rhizome	Peat	B	v	Without rhizome	Peat	B
k	With rhizome	Peat	C	w	Without rhizome	Peat	C
l	With rhizome	Peat	D	x	Without rhizome	Peat	D

**Table 3 plants-14-00744-t003:** Interaction code and corresponding evaluation factors (shoots and substrate) for the different propagation parameters of *A. queko*.

Code Interaction (X Axis—Figure 6)	Shoots	Substrate
a	With rhizome	Peat
b	With rhizome	Mulch
c	With rhizome	Black soil
d	Without rhizome	Peat
e	Without rhizome	Mulch
f	Without rhizome	Black soil

## Data Availability

Data can be found at https://drive.google.com/drive/folders/1Tu4Ox3nWXCUnBTRTlmaSJsAX6x64PaHz.

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
