# Peer review of "Ecological Relationships Between Woody Species Diversity and Propagation Strategies of Aulonemia queko"

_plants, 2025, doi:10.3390/plants14050744_

Round 1
Reviewer 1 Report
Comments and Suggestions for Authors
Major Comments
- Line 13-29: The abstract lacks a critical evaluation of the findings. Instead of only stating the results, provide a discussion on the broader ecological implications and potential limitations. How do these findings contribute to conservation strategies for A. queko?
- Line 40-46: The introduction mentions that bamboo dominance influences montane forest composition, but it does not provide clear mechanisms. How does A. queko specifically alter nutrient cycling, light availability, or competition dynamics? Adding references or hypotheses would strengthen this section.
- Line 66-77: The rationale for selecting rhizome-based propagation and IBA hormone treatments is not well justified. Are these methods based on previous studies of A. queko, or were they inferred from general bamboo propagation techniques? A brief review of relevant literature would be helpful.
- Line 82-84: The PERMANOVA analysis suggests significant variation in species composition, but the NMDS results are not clearly explained. What percentage of variation is explained by the NMDS axes, and how strong are these differences ecologically? A more detailed interpretation is needed.
- Line 101-104: The correlation results indicate that A. queko dominance explains floristic composition variations, but the biological mechanisms underlying this influence are not discussed. Could this be due to allelopathic effects, competition, or microhabitat modification?
- Line 109-112: The survival analysis of A. queko shoots suggests that rhizome-bearing shoots have a higher success rate, but the lack of significance for substrate and hormone treatments contradicts expectations. Could this be due to sample size limitations, or are there ecological factors at play? Consider discussing this more critically.
- Line 121-128: The growth response to mulch and IBA at 1000 ppm is interesting but lacks discussion on why mulch might enhance growth. Is it due to water retention, nutrient content, or microbial interactions? A mechanistic explanation is needed.
- Line 149-153: The finding that SLA is higher in shoots without rhizomes is surprising, as one would expect rhizomes to enhance overall vigor. Could this be due to stress-induced morphological adaptations? Discuss alternative interpretations.
- Line 183-197: The claim that A. queko's dominance influences forest composition should be supported by additional ecological theories. How does this compare to other bamboo species' ecological roles in tropical forests? A broader context would be beneficial.
- Line 219-227: The interaction effects between substrate, rhizomes, and hormone doses suggest a complex physiological response. Have similar findings been reported in other bamboo species? If so, compare your results to existing studies for validation.
- Minor Comments: Line 23: The phrase "demonstrating their interdependence" is vague. Specify which factors showed the strongest interactions.
-
Line 45: The term "adaptive functional traits" is used, but no examples are provided. Consider specifying traits such as root-to-shoot ratio, wood density, or leaf morphology.
Line 87: The NMDS plot is referenced, but no stress value is provided. Report the stress value to assess the reliability of ordination results.
Line 99: The term "dominance" is used frequently, but it is unclear whether it refers to relative abundance, basal area, or another metric. Define this term more precisely.
Line 134: Table 2 contains interaction codes but lacks descriptions of statistical significance. Consider adding p-values or effect sizes for transparency.
Line 144: Figure 5 presents mean values but does not include confidence intervals. Adding error bars would improve data interpretation.
Line 168: The discussion states that elevation influences floristic composition, but the effect size is not provided. Quantifying this variation would add rigor.
Line 211: The statement that rhizomes provide energy reserves for new shoots should be supported with citations from bamboo physiology literature.
Line 236: The role of auxins in root development is briefly mentioned. Consider expanding on how auxins influence lateral root initiation in monocots versus dicots.
Line 398: The conclusion states that A. queko plays a "pivotal role" in Andean forests, but this is not strongly justified. Are there quantitative measures of its ecological impact compared to other dominant species?
The manuscript is generally well-structured, but there are several instances of awkward phrasing, grammatical errors, and unclear sentences that affect readability.
Reviewer 2 Report
Comments and Suggestions for Authors
Ecological relationships between woody species diversity and propagation strategies of Aulonemia queko
This is a well-written and well-designed paper that will be an excellent and high-quality contribution to our understanding of Aulonemia queko. The experimental design is valid, and the results of the experiment are worthy of publication. The use of English is superb. I recommend accepting the paper with only a few suggested edits, high-lighted in red throughout the paper.
The use of “belt” and “belts” should be changed to be “zone” and “zonation” (referring to plant zonation and plant zones) throughout the paper. Through conventional uses, sentences that begin with a scientific genus such as Aulonemia must fully spell out the name and can’t abbreiviate it – though it obviously can otherwise (so Aulonemia not A. at the beginning of sentences).
Early in the paper, in the Introduction at lines 11-12, for example, it would be useful to have a reference through which the taxonomy and a botanical description of Aulonemia queko can be found. I suggest the Kew Royal Botanic Gardens “Plants of the World” or something similar to it. I would recommend adding it as around reference number 13 or in that vicinity of papers cited early in the paper.
Kew Royal Botanic Gardens. Plants of the World.
https://powo.science.kew.org/taxon/urn:lsid:ipni.org:names:26916-2#higher-classification
Placing the Results section so early in the paper seems unique to me, as I am used to seeing the conventional approach of having them much later in the paper (certainly after Materials and Methods). I prefer the traditional placement, so I urge the authors to consider placing the Results section in the later part of the paper.
The photographs in Figure 8 are excellent and clearly show the features the illustrate.
The authors should make sure that the doi links work in all of their references – or don’t include the links for those (or find the correct one). Links didn’t seem to appear for me when I tried to go to them in Reference numbers 5, 6, 22, 23, and 24.
I urge the authors to consider the suggested edits that are highlighted in red, in this otherwise publishable paper.

Author Response
Dear Reviewer,
We appreciate your time and effort in evaluating our manuscript. Additionally, we are grateful for your valuable suggestions, which have helped improve the document.
Regarding your comments, we have addressed all of them. Specifically, we have replaced "belt" with "zone," incorporated your suggested edits, and included the recommended bibliographic citations. You can find all these changes in the revised version of the manuscript, which we have submitted to the Editor with tracked changes to highlight our corrections.
In this document, you will also find additional modifications corresponding to the corrections that another reviewer suggested.
Sincerely,
The Authors
Round 2
Reviewer 1 Report
Comments and Suggestions for Authors
In the methods section, provide more detail on the statistical methods used, particularly for the PERMANOVA analysis. Clarifying assumptions and criteria for significance could enhance reproducibility.
In the discussion, consider elaborating on the implications of the findings for conservation practices. More specific recommendations based on the results could strengthen the manuscript.
Comments on the Quality of English LanguageA thorough proofreading for typographical errors and grammatical inconsistencies should be conducted. For instance, check the consistency of terms like "bamboo" and "bamboos," and ensure that scientific names are italicized correctly.